# TTCD: TRANSFORMER INTEGRATED TEMPORAL CAUSAL DISCOVERY FROM NON-STATIONARY TIME SERIES DATA

## ABSTRACT

The widespread availability of complex time series data in various domains such as environmental science, epidemiology, and economics demands robust causal discovery methods that can identify intricate contemporaneous and lagged relationships in non-stationary, nonlinear, and noisy settings. Existing constraint-based methods often rely heavily on conditional independence tests that degrade for limited data samples and complex distributions, while score-based methods impose strong statistical assumptions. Recent methods address special cases such as change point detection or distribution shifts, but struggle to provide a unified solution. We propose the **T**ransformer Integrated **T**emporal **C**ausal **D**iscovery (TTCD) Framework, a novel end-to-end approach that learns contemporaneous and lagged causal relations from non-stationary time series. TTCD introduces a Non-Stationary Feature Learner integrating temporal and frequency-domain attention with dynamic non-stationarity profiling, and a custom Causal Structure Learner. A key innovation is reconstruction-guided causal signal distillation, to distill essential causal signals through the reconstruction process of the transformer decoder, which mitigates noise and spurious correlations while preserving meaningful dependencies. The Causal Structure Learner operates on distilled reconstructed signals to infer the underlying causal graph without restrictive assumptions on noise distributions or data generation processes. Experiments on synthetic, benchmark, and real world datasets show that TTCD consistently outperforms state-of-the-art baselines in both accuracy and consistency with domain knowledge, demonstrating the approach's effectiveness for causal discovery in challenging real world contexts.

## 1 INTRODUCTION

Time series generated by natural systems such as climate, finance, economics, and healthcare often exhibit non-linearity, non-stationarity, different noise types, and autocorrelation (Runge et al., 2019a). These intricate properties pose significant challenges for understanding dependencies among system components. A common approach to simplify this complexity is to graphically represent the data generation model using directed acyclic graphs (DAGs), which is a very convenient way to express complex systems in a highly interpretable manner and also provides causal insight into the underlying processes (Pearl, 2000). DAG representation of a system plays a vital role in decision making and prediction of future conditions in different applications such as causal inference (Pearl, 1991; Spirtes et al., 2000), neuroscience (Rajapakse & Zhou, 2007), medicine (Heckerman et al., 1992), economics (Appiah, 2018; Sanford & Moosa, 2012), etc. However, learning DAGs from observational time series data is very challenging when controlled experiments with different population sub-groups are impractical or unethical (Spirtes et al., 2000; Peters et al., 2017).

Several state-of-the-art methods have been developed for causal discovery from temporal data based on constraint-based and score-based methodologies. Constraint-based methods (Runge et al., 2019b; Runge, 2020; Gerhardus & Runge, 2020; Entner & Hoyer, 2010; Huang et al., 2020) learn conditional independencies through statistical tests to build DAGs. However, conditional independence tests (CIT) require a large number of samples to generate reliable test scores and can struggle with complex data distributions, often generating equivalence classes instead of precise causal graphs

(Shah & Peters, 2020; Huang et al., 2018; Glymour et al., 2019). Errors in the early stage can be impacted by cascading errors in later stages, and CIT at multiple stages can lead to false detection results (Li et al., 2019; Triantafillou & Tsamardinos, 2016).

Score-based causal discovery methods use a score function to quantify a predicted causal graph and optimize it gradually by enforcing the acyclicity constraint (Glymour et al., 2019; Huang et al., 2018; Triantafillou & Tsamardinos, 2016). By evaluating the entire graph instead of applying sequential tests, they mitigate error propagation and multi-stage inconsistencies. However, the large combinatorial search space of an adjacency matrix makes this optimization challenging and often requires additional DAG constraints. Zheng et al. (2018) transform this combinatorial problem into a continuous optimization by formulating an acyclicity constraint using the trace exponential of the predicted adjacency matrix, enabling gradient-based optimization. Based on this, several neural network-based methods have been proposed (Zheng et al., 2020; Sun et al., 2023; Pamfil et al., 2020; Yu et al., 2019; Löwe et al., 2022). But these methods often face the overfitting issue due to noise or spurious correlations in small-sample settings, and most methods assume stationarity. Recently, transformer architectures have also been explored to analyze time series data (Wen et al., 2023; Zeng et al., 2023; Kong et al., 2024).

Causal discovery from non-stationary temporal data remains an active research area (Gong et al., 2024) and several advanced methods have been proposed in constraint-based (Ferdous et al., 2023; Zhifeng et al., 2024; Sadeghi et al., 2024) and score-based (Schäck et al., 2017; Liu & Kuang, 2023; Mameche et al., 2025; Rodas et al., 2021) categories for this task. However, these methods are designed to address specific scenarios such as change point detection, shift in data distribution, conditional stationarity, change in causal relationships, or summary graphs. Some existing approaches also require prior knowledge of noise distribution and parametric information of data generation. Therefore, in this paper, we propose a causal discovery framework capable of capturing causal structure from non-stationary temporal data without any noise or data distribution assumptions. Our proposed framework integrates a transformer-based non-stationary feature learner with a custom 2D convolution to capture causal relationships between each variable and its temporal parents. The contributions of this paper are three-fold:

- We propose a non-stationary transformer to learn dominant features from time series data using both temporal and frequency domain attentions with non-stationary profiling and de-stationary feature learning, which provides specific attention on important features.

- We propose a convolution-based Causal Structure Learner to learn the causal relationships from distilled signals. The proposed module can identify lagged and contemporaneous causal links simultaneously using the acyclicity constraint and sparsity penalty into the optimization process.

- We conduct extensive evaluations of the proposed framework with state-of-the-art causal discovery methods and ablation studies using synthetic and real world datasets. The proposed framework performs better than state-of-the-art approaches in most cases, making it a strong contender for time-series causal discovery.

## 2 RELATED WORKS

Traditional statistical causal discovery methods were not designed to handle non-linear data. While some methods extend the traditional causal discovery methods to handle non-linear time series data, such as PCMCI and PCMCI+ (Runge et al., 2019b; Runge, 2020; Bahadori & Liu, 2012), some approaches utilize neural networks for these extensions (Yu et al., 2019; Tank et al., 2021; Absar et al., 2023; Zheng et al., 2020; Pamfil et al., 2020; Sun et al., 2023). For instance, DAG-GNN (Yu et al., 2019) leverages neural networks and gradient-based optimization to identify causal structures.

Recent research has made inroads to propose causal discovery techniques applicable to non-stationary time-series data, constraint-based methods (Huang et al., 2020; Sadeghi et al., 2024; Ferdous et al., 2023; Zhifeng et al., 2024) and score-based methods (Rodas et al., 2021; Schäck et al., 2017; Liu & Kuang, 2023; Mameche et al., 2025). Causal Discovery from NOnstationary Data (CD-NOD) (Huang et al., 2020) is a nonparametric framework that identifies causal relations from non-stationary data based on distribution shift. Causal Discovery from Nonstationary Time Series (CD-NOTS) extends CD-NOD to find lagged and instantaneous causal links using CITs (Sadeghi

et al., 2024). Ferdous et al. (2023) proposed CDANs, which reduces the conditioning set by considering lagged parents and utilizes changing modules to detect causal edges. Zhifeng et al. (2024) introduced a causal discovery method that divides the time series into several stationary intervals using a change detection method and applies a stationary method to individual intervals.

Score-based method, State-Dependent Causal Inference (SDCI) (Rodas et al., 2021) assumes the dynamics of a non-stationary system change based on different states, and conditioning on each state applies a probabilistic deep learning approach to learn causal graphs. Schäck et al. (2017) proposed a method by integrating a time-varying autoregressive method and generalized partial directed coherence (PDC), where the Kalman filter is used to predict PDC parameters. Latent Intervened Nonstationary learning (LIN) (Liu & Kuang, 2023) method assumes data contains both observational and interventional samples, learns causal graphs for each class using a neural network and acyclicity constraint. SPACETIME (Mameche et al., 2025) method considers changes in time and space simultaneously to detect causal graphs from multi-context data using Gaussian processes. Fujiwara et al. (2023) combined Linear Non-Gaussian Acyclic Model (LiNGAM) and the Just-In-Time (JIT) framework to identify causal relations in nonlinear and non-stationary data.

While these methods have significantly contributed to the field of non-stationary time series causal discovery, several challenges persist. The constraint-based methods highly rely on conditional independence tests and are prone to error propagation. Also, multi-stage tests can lead to an increased risk of false positives or negatives. Though recent score-based causal discovery methods for non-stationary temporal data mitigate these issues to some extent, they deal with specific challenges, like context change, distribution shift, interventional data, conditional and local stationary, etc. Natural non-stationary temporal data does not match these criteria for all cases. By relaxing specific conditions on data distribution and data generation mechanism, our proposed framework learns non-stationary features from natural temporal data and generates effective temporal causal graphs. The comparison between different existing methods is provided in Appendix A.

## 3 PRELIMINARIES

Let's consider a multivariate time series dataset $X = \{x^1, x^2, x^3, \ldots, x^n\}$ consisting of $n$ variables, and each variable is measured for $T$ timesteps. Variable $x^i(i \in \{1, ..., n\})$ at a specific time point $t \in T$ could be caused by other variables at the same timestep $(t)$ and all variables from previous timesteps (0 to $t - 1$), following the temporal precedence assumption (the output causal graph of Figure 1). The effects from previous timesteps, also called lagged effects, can propagate from infinite earlier time points, but for DAG learning purposes, we will consider a maximum time lag, i.e., $l_{max}$.

**Definition 1:** Consider a time series $X_t = (X_t^i)_{i \in \{1,...,n\}}$ with continuous distribution. If there is a $l_{max} > 0$ and $\forall i \in n$ there are sets $PA_t^{x^i} \subseteq X_t^{n \setminus i}$, $PA_{0...(t-1)}^{x^i} \subseteq X_{0...(t-1)}$, the structural equation model is

$$X_t^i = f_i(PA_{t-l_{max}}^{x^i}, ..., PA_{t-1}^{x^i}, PA_t^{x^i}, e_t^i), \tag{1}$$

with noise term $e_t^i$. So the set of possible cause variables of each time series $x^i$ at time $t$ is $PA^{x^i} \in [\{X_{(t-l_{max})}, X_{(t-l_{max}+1)}, ..., X_{(t-1)}, X_t\} - x^i]$. The goal is to learn a causal graph $G(V, E)$ such that its vertices resemble time-lagged and current time variables, and its directed edges express causal links. So the vertices and edges can be denoted as $V = \{X_{(t-l_{max})}, X_{(t-l_{max}+1)}, ..., X_{(t-1)}, X_t\}$, $E = \{(V_i, V_j) : V_i, V_j \in \{X_{(t-l_{max})}, X_{(t-l_{max}+1)}, ..., X_{(t-1)}, X_t\}\}$, respectively. Let the weighted adjacency matrix of full temporal causal graph G be denoted by $W \in \mathbb{R}^{(n \times (l_{max}+1)) \times n}$.

The proposed method works based on the following assumptions. Markov and Faithfulness Assumption: Assume $P^{(X^i)}, i \in \{1, ..., n\}$ is Markov and faithful to the true/generated causal graph $G$ (Hasan et al., 2023). Causal Sufficiency Assumption: We assume that there are no hidden/unobserved confounders in the data generation process. Causal Consistency Assumption: Assume causal relations between the variables are consistent through all time steps. Acyclicity Assumption: This assumption states that there are no causal paths that begin and end at the same node. Assuming temporal precedence in the data ensures acyclicity constraint in the time-lagged part of $W$. However, for the contemporary part of $W$ at $t$, each node can serve as both the source and target of causal links, required to maintain the acyclicity. Simultaneously learning the lagged and

contemporaneous parts of the adjacency matrix is very challenging for complex datasets. Since any variable might be the cause of another effect variable, cycles can occur in the contemporaneous part of the adjacency matrix.

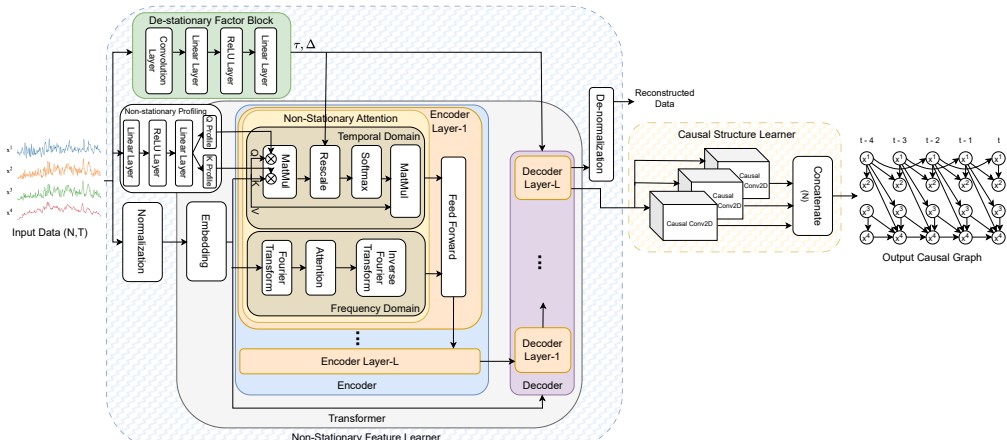

Figure 1: Proposed TTCD framework to learn full temporal causal graph. The Non-Stationary Feature Learner module learns distilled reconstructed features using temporal and frequency domain attentions, a non-stationary profiling network, and a de-stationary factor block. The Causal Structure Learner operates on distilled reconstructed signals to generate a full causal graph.

# 4 PROPOSED METHODOLOGY

The causal graph generating task can be treated as an unsupervised learning process of the adjacency matrix $W$ given $T$ observations of a multivariate time series data $X$. To learn a directed adjacency matrix $W$ of the full temporal causal graph $G$, we propose a framework based on unsupervised deep neural networks. The proposed framework learns the instantaneous ($X_t \rightarrow X_t$) and time-lagged ($\{X_{(t-l_{max})}, X_{(t-l_{max}+1)}, ..., X_{(t-1)}\} \rightarrow X_t$) causal links for maximum time lag ($l_{max} > 0$). Our proposed framework is illustrated in Figure 1, which consists of two modules: a Non-Stationary Feature Learner module and a Causal Structure Learner module. Non-Stationary Feature Learner leverages a transformer-based encoder with a specialized attention mechanism to learn latent representations from temporal and frequency domain features. The decoder reconstructs the input signal from these latent representations, and we use the reconstructed output (prior to denormalization) as input to the causal structure learner. This design serves two purposes: it ensures dimensional alignment with the original data space, and it filters out spurious fluctuations and noise while retaining meaningful dynamics. The Causal Structure Learner module contains $N$ nonsequential custom causal layers (Causal Conv2D) to learn time-lagged and instantaneous causal relationships of each input variable to its parent variables. Following a similar analogy used by Zheng et al. (2020) and DAG-GNN (Yu et al., 2019), we will learn causal links from parameters of Causal Conv2D layers. Here, the links learned by this module are always unidirectional. The unique design of this module helps to learn the causal links for each input variable independently. Finally, the results of each causal layer are aggregated to generate a causal graph of the input data. Causal graph identifiability of the proposed model is inspired by Peters et al. (2013; 2011) and follows a similar behavior. Please refer to Appendix B for details.

## 4.1 NON-STATIONARY FEATURE LEARNER

The Non-Stationary Feature Learner module of our proposed framework learns latent representation from input time series data leveraging non-stationary attention introduced by Liu et al. (2022). Motivated by the boosted performance of their model, we follow a similar transformer strategy to learn features from non-stationary temporal data. The input data is divided into sequential chunks ($In \in \mathbb{R}^{(T \times (l_{max}+1) \times n)}$) to maintain inherent temporal order, and an embedding is generated for each chunk ($E \in \mathbb{R}^{(T \times (l_{max}+1) \times d_e)}$). Encoder block of the transformer takes this embedding as input to compute attention scores using non-stationary attention and learns the latent representation.

The non-stationary transformer can learn important features of input time series; however, the non-stationarity cannot be fully covered through attention only, because the data normalization process attenuates non-stationary characteristics. Due to normalization, sequences from distinct time series can appear statistically identical, causing the model to generate uniform attention without being aware of important features. This ignorance of crucial non-stationary features limits the quality of the learned features and weakens the overall performance. To tackle this problem, we explicitly derive non-stationary components from raw input and integrate them into attention computation such that the transformer can retain significant non-stationarity in its representations. We achieve this by introducing a Non-Stationary Profiling network and a de-stationarizing module, De-Stationary factor learning Block (DSB). Specifically, the Non-Stationary Profiling Network extracts dynamic, localized statistics—such as local variability or higher-order moments—capturing sample-specific distributional profiles that are often suppressed by standard normalization. These learned profile vectors ($\gamma_Q^{(i)}$, $\gamma_K^{(i)} \in \mathbb{R}^{T \times d_e}$) work as meta-conditioning signals that adapt the transformer's attention weights for each input dynamically. This makes the transformer data-adaptive, not just parameter-adaptive, and goes beyond fixed decomposition.

$$Q^{(i)} = Q \odot \gamma_Q^{(i)}, K^{(i)} = K \odot \gamma_K^{(i)}$$
$$where \big[\gamma_Q^{(i)}, \gamma_K^{(i)}\big] = Profile\big(X^{(i)}\big) \tag{2}$$

The De-Stationary Factor Learning Module (DSB) is designed to explicitly restore and amplify the intrinsic non-stationary characteristics of time series data, which are often attenuated or lost due to standard normalization and sequence embedding steps. In our framework, the non-stationary profiling network focuses on local features, and the DSB captures broader, global non-stationary profiles. As shown in Figure 1, the DSB comprises a convolution layer and several linear layers with ReLU activations. This block learns scaling factor $\tau \in \mathbb{R}^{T \times 1}$ (equivalent to $\sigma^2$) and shifting vector $\Delta \in \mathbb{R}^{T \times (l_{max}+1)}$ (equivalent to $\mu$) utilizing raw input data and its computed mean $\mu_x$ and standard deviation $\sigma_x$. These learned de-stationary factors are then integrated with the attention computing mechanism (Equation 3) to learn varying attention considering non-stationarity.

$$Attn(Q, K, V, \tau, \Delta) = Softmax(\frac{\tau QK^\top + I\Delta^\top}{\sqrt{d_e}})V, \tag{3}$$

where Q, K and V represent the query, key and value matrices of the transformer with dimension $\mathbb{R}^{T \times (l_{max}+1) \times d_e}$, respectively, and $I$ is a vector of all ones. These learned de-stationary factors are used inside the attention module to multiply learned attention values. Additionally, these learned de-stationary factors are shared by all attention modules used in the whole transformer.

Recent studies by Zhou et al. (2022), Yi et al. (2023), and Li et al. (2025) have demonstrated that integrating frequency domain attention significantly improves the model's capacity to disentangle non-stationary time series and identify latent causal drivers. So, we integrate a Frequency Domain Attention alongside the standard temporal attention to further enhance the model's ability to capture complex non-stationary patterns. A Fourier Transform is applied to convert time-domain signals into frequency spectra ($Fre \in \mathbb{R}^{(T \times ((l_{max}+1)/2) \times d_e)}$), enabling the network to selectively attend to distinct spectral bands and periodic components of non-stationarity in the signal. This frequency domain attention is fused with the time domain latent features conditioned by local profile vectors and de-stationary factors, enabling the model to simultaneously exploit localized distribution and frequency-based dependencies. This multiview representation enhances the learner's ability to detect time-lagged and instantaneous causal links that are modulated by complex non-stationary dynamics. The learned latent representation ($\mathbb{R}^{T \times (l_{max}+1) \times d_e}$) of the encoder module is provided as input to the transformer decoder module together with generated input data embeddings. The decoder module also uses non-stationary attention blocks with de-stationary factors to reconstruct input data ($\mathbb{R}^{T \times (l_{max}+1) \times n}$). The distilled signals learned by the decoder module are provided as input to the proposed Causal Structure Learner module to learn causal relationships. The output of the decoder module is still on an unnormalized scale. Therefore, the de-normalization block is used to shift the output back to the original scale of the input data.

## 4.2 CAUSAL STRUCTURE LEARNER

We propose this novel module to learn lagged and instantaneous causal links of each variable using distilled reconstructed signals ($\mathbb{R}^{T \times (l_{max}+1) \times n}$) learned by the decoder block. This consists of a separate custom Causal Conv2D layer for each variable, and these layers are organized in a non-sequential pattern. The Causal Conv2D layer takes input in a similar structure as shown in the full

causal graph on the right side of Figure 1, with lagged data followed by data from the current time point. Each Causal Conv2D layer is designed to learn causal links of an input variable, for example, to $x^1$ from all possible parents $PA^{x^1} \in \{x^1_{(t-l_{max})}, x^1_{(t-l_{max}+1)}, ..., x^1_{(t-1)}, x^2_{(t-l_{max})}, x^2_{(t-l_{max}+1)}, ..., x^2_{(t-1)}, x^2_t, ..., x^n_{(t-l_{max})}, x^n_{(t-l_{max}+1)}, ..., x^n_{(t-1)}, x^n_t\}$ of that variable following Definition-1. The variable itself cannot be included in the set of its parent variables. Let's assume we have a time series dataset with 4 variables, and for lagged effects, consider a maximum time lag $l_{max} = 4$. So, the input data size will be $(4 \times 5)$, one row for each variable and $l_{max} + 1 = 5$ columns for lagged and contemporaneous data. To learn a causal graph for 4 variables, as shown in Figure 2, we have to employ 4 Causal Conv2D layers. Each of these layers predicts the expectation of a target variable in the current timestep $t$ given all lagged and instantaneous parents (Equation 4).

$$E[x^i | PA^{x^i}] = f_{W^{x^i}}(PA^{x^i}) \tag{4}$$

Here, $f_{W^{x^i}}()$ denotes the function learned for target variable $x^i$ and $W^{x^i}$ represents the set of weight parameters of that layer. The adjacency matrix is derived from learned weights of these Causal Conv2D layers. Each weight parameter of a layer related to a target variable represents the strength of causal links from its potential parent. If a weight parameter $W^{x^k}_{ij} = 0$, this means that the target variable $x^k$ is independent of the cause variable $x^i$ at timestep $j$. Conversely, if $W^{x^k}_{ij} > 0$, the target variable $x^k$ has a causal edge from parent variable $x^i$ at a specific time lag $j$. After training weight parameters of all target variables, we apply a thresholding operation to prune causal links with weak dependency strength, $W^{x^k}_\omega = (W^{x^k} > \omega)$, where $\omega$ is a minimum threshold limit. After thresholding, the weight parameters of all variables represent the adjacency matrix of the generated causal graph.

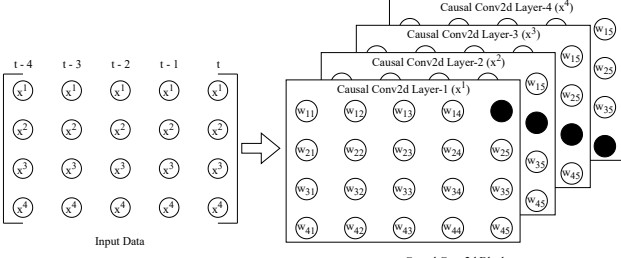

Figure 2: Example of the proposed custom causal layers with four variables and a time lag of 4.

### 4.3 Optimization

To train the proposed framework, we optimize these four terms in the objective function: transformer reconstruction loss, target variable estimation loss, acyclicity constraint, and sparsity loss. *Reconstruction loss*: In the Non-Stationary Feature Learner module, we learn latent representations and reconstruct them to the input data through the transformer's encoder-decoder structure. Therefore, to optimize this module, we use the mean squared error (MSE) loss of input data $X$ and reconstructed output $\widehat{X}$, which is defined as:

$$L_r = \frac{1}{T} \sum_{i=1}^{T} \left\| X - \widehat{X} \right\|_2^2 \tag{5}$$

*Acyclicity Constraint*: To optimize the Causal Structure Learner module we have to ensure acyclicity property of the causal graph. To enforce acyclicity in the adjacency matrix of the learned causal graph, we use an equality constraint similar to that of Zheng et al. (2018), formulated as $h(W) = 0$. The function $h(W)$ is defined using the trace exponential ($tr$) of the elementwise product of the adjacency matrix with itself. $h(W) = tr\,e^{W \odot W} - n$, here $n$ is the number of variables. We cannot use the learned adjacency matrix $W$ directly in this equality function because $W$ contains both time-lagged $(t-l_{max}, t-l_{max}+1, \ldots, t-1)$ and contemporaneous $(t)$ edges of the causal graph. Lagged causal edges always redirect from previous timesteps to the current timestep $t$, and are acyclic. We have to apply acyclicity constraint only to the contemporaneous part of the adjacency matrix $W^t$.

$$h(W^t) = tr(e^{W^t \odot W^t}) - n = 0 \tag{6}$$

The function $h(W^t)$ will be equal to 0 if and only if the corresponding matrix $W^t$ does not have any cycle. However, we cannot directly integrate this equality constraint into a continuous optimization

framework. But the equality constraint $h(W^t) = 0$ can be solved using continuous optimization after converting this into an unconstrained problem (Zheng et al., 2018). Therefore, we transform this equality constraint into an unconstrained subproblem using the augmented Lagrangian method. The continuous form of this constraint is defined as:

$$h(W^t) = 0 \approx \min \left[ \frac{\rho}{2} |h(W^t)|^2 + \alpha h(W^t) \right],$$ (7)

where $\alpha > 0$ is the Lagrange multiplier, $\rho > 0$ is the penalty parameter of the augmented Lagrangian.

*Target Variable Estimation Loss*: The proposed Causal Structure Learner module estimates each target variable by using all possible parents to learn the adjacency matrix $W \in R^{(n \times (l_{max}+1)) \times n}$ of the desired causal graph. To improve this estimation quality, we must consider the difference between estimated and actual values of target variables. Therefore, we define a mean squared error loss ($L_{tve}$) between true values and estimated values obtained through causal connections:

$$L_{tve}(W) = \frac{1}{T} ||X - WX||_F^2$$ (8)

*Sparsity Loss*: We incorporate an additional penalty to enforce sparsity in the learned adjacency matrix using the $L1$ norm of $W \in R^{(n \times (l_{max}+1)) \times n}$. This penalty term helps to generate fewer causal links with strong relationships. While $L_r$, $L_{tve}$, and the acyclicity constraint (Equation 7) drive the model to find dense and overly connected relationships to minimize loss, potentially increasing the number of non-zero weights—the $L1$ norm of $W$ try to reduce less significant weights to zero to keep a minimum number of non-zero entries. The sparsity loss is defined as $L_s = \lambda ||W||_1$, where $\lambda$ is the sparsity penalty regularization. By combining all loss terms, the objective function of the proposed framework becomes:

$$\min_W \left[ L_r + L_{tve}(W) + \frac{\rho}{2} |h(W^t)|^2 + \alpha h(W^t) + L_s \right],$$ (9)

where the 3rd and 4th terms are augmented Lagrangian for the acyclicity constraint. This objective function can be minimized using any state-of-the-art continuous optimizer.

## 5 EXPERIMENTAL SETUP

We describe datasets and evaluation metrics used for performance comparison in this section. Our model is developed using the PyTorch library, and all experiments are conducted on Google Colab Runtime with CPU for easy reproducibility. A fixed random seed value is used for randomized data to make the experimental results reproducible. The implementation code and datasets used for this study are available at https://anonymous.4open.science/r/TTCD/README.md.

**Synthetic Datasets:** We used two synthetic datasets to evaluate the performance of our proposed causal discovery method. As we know the ground truth causal graph for synthetic datasets, we can measure and compare generated causal graphs easily. We generated a time series dataset (Dataset-1) consisting of four variables using Gaussian white noise $\varepsilon$ following a similar data generation process presented in (Huang et al., 2020), which contains both lagged and instantaneous links. A mathematical description and the true causal graph for this dataset are provided in Appendix C. Non-stationary characteristic is incorporated into the generation process of this dataset to mimic the dynamic properties of real world natural system.

The other synthetic dataset (Dataset-2) follows a similar data generation process presented in (Kang et al., 2022). For this dataset, we used exponential nonlinearity and noise signals from the Poisson distribution. The mathematical equations of this dataset also given in Appendix C. All the variables of this dataset are also non-stationary.

**Real World Datasets:** Two real world Earth/atmospheric science datasets, namely Turbulence Kinetic Energy (TKE) and Arctic Sea Ice, and FMRI benchmark data were used to evaluate our work. These natural datasets exhibit high variability, non-stationarity, and complex interactions. TKE refers to the mean kinetic energy per unit mass of eddies in turbulent flow (Hinze, 1975). The temporal TKE data used in this study represent the TKE evolution during a typical cumulus-topped boundary layer day (local time 05:00–18:00) over the DOE Atmospheric Radiation Measurement (ARM) Southern Great Plains Central Facility. This data file is generated from an idealized numerical simulation using the Weather Research & Forecasting Model (Skamarock et al., 2019) with modifications

from the Large-Eddy Simulation (LES) Symbiotic Simulation and Observation (LASSO) activity, which is developed through the US Department of Energy's ARM facility (Gustafson et al., 2020; Endo et al., 2015). The dataset also includes TKE vertical shear production ($SH$) and buoyancy production ($BU$) terms, which together yield the net TKE tendency ($TEND$). Their ground-truth relationships are shown in Figure 3b (Appendix C), and non-stationarity test results are available in Appendix E.

Arctic sea ice is an important component of the world's climate system and plays a significant role in the rise of extreme weather events. Huang et al. (2021) conducted a causal discovery analysis to investigate the links between the melting Arctic sea ice and atmospheric variables. We use the same 11 atmospheric variables with the sea ice extent employed in (Huang et al., 2021). This time series data contains monthly averages from 1980 to 2018 over the Arctic region north of 60N. The variable names and non-stationary test results for this dataset are provided in Appendix D and E respectively.

The FMRI benchmark dataset (Smith et al., 2011) provides rich, realistic simulated blood-oxygen-level-dependent (BOLD) time series for modeling brain networks. Activity between 5 brain regions is measured in this dataset, considering the change in blood flow. Each brain region is considered a node and 2400 samples are recorded for each node. The ground truth causal graph of this dataset is also provided for performance evaluation.

**Evaluation Metrics:** We evaluate the performance of our proposed causal discovery method using Structural Hamming Distance (SHD), F1 Score and False Discovery Rate (FDR). SHD represents the number of edge corrections (deletion, insertion) to match the predicted graph with the true causal graph. FDR explains the rate of predicted wrong edges from all predicted edges considering the direction of each edge. F1 Score calculates the harmonic mean of precision and recall. The F1 score ranges from 0 to 1 and a higher value means a better prediction of the true graph. In contrast, lower SHD and FDR represent better performance of the causal discovery method.

## 6 RESULTS

In this section, we present the comparative results of time series causal discovery between the proposed method and state-of-the-art methods. To evaluate the performance of our proposed method, we considered 8 SOTA methods as baselines: CD-NOD (Huang et al., 2020), LIN (Liu & Kuang, 2023), PCMCI+ (Runge, 2020), DYNOTEARS (Pamfil et al., 2020), NTS-NOTEARS (Sun et al., 2023), PCMCI (Runge et al., 2019b), NOTEARS-MLP (Zheng et al., 2020), and DAG-GNN (Yu et al., 2019). The first six methods can learn causal graphs for time series data; among these, CD-NOD (Huang et al., 2020) and LIN (Liu & Kuang, 2023) work on non-stationary data. Although the LIN method assumes both intervention and observation samples, in our experiments, we set the intervention parameter to 0 to model observation data, which means no intervention is applied to the data. While the NOTEARS-MLP (Zheng et al., 2020), and DAG-GNN (Yu et al., 2019) methods were proposed for non-temporal data, we include these methods due to their strong performance and widespread usage in different domains (Entner & Hoyer, 2010; Huang et al., 2021). We transformed the lagged and instantaneous data into a long sequence such as $\{x_{t-5}^1, x_{t-5}^2, x_{t-5}^3, x_{t-5}^4, x_{t-4}^1, x_{t-4}^2, x_{t-4}^3, x_{t-4}^4, \ldots, x_t^1, x_t^2, x_t^3, x_t^4\}$, so that we could apply transformed dataset to the non-temporal methods to find lagged and current time causal relationships. For fair comparison, we carefully tuned hyperparameters for each method to get the best evaluation scores. The hyperparameters used for all baseline methods are provided in Appendix F.

To evaluate the performance of the baseline methods, we compared the predicted causal graph in the full temporal graph setting, considering both edge direction and time lag of each edge. The qualitative comparison is reported in Table 1, where the best scores are marked in bold, and underlined values represent the second best scores. From Table 1, we can see that our proposed method obtained the best results on Synthetic Dataset-1, TKE, and Arctic Sea Ice datasets, and comparable results for other datasets. For Dataset-2, the TCDF method yielded the same SHD and FDR scores but a lower F1 score, indicating this method predicted fewer edges than the ground truth edges. For the FMRI dataset, DAG-GNN method achieved the best F1 score, where the proposed method generated a better FDR score with the same SHD. As this dataset represents chain-like relationships, the proposed method failed to detect all target edges, eventually generating fewer edges with high causal strengths. Baseline methods for non-stationary data also performed well on FMRI dataset. Overall, these comparative results demonstrate that our proposed framework is capable of generating better

Table 1: Comparative analysis of the full causal graph predicted by different baseline methods for synthetic and real world datasets.

| METHOD | DATASET-1 | | | DATASET-2 | | | TKE | | | ARCTIC SEA ICE | | | FMRI | | |
|---|---|---|---|---|---|---|---|---|---|---|---|---|---|---|---|
| | SHD↓ | F1↑ | FDR↓ | SHD↓ | F1↑ | FDR↓ | SHD↓ | F1↑ | FDR↓ | SHD↓ | F1↑ | FDR↓ | SHD↓ | F1↑ | FDR↓ |
| PCMCI | 24 | 0.36 | 0.75 | 36 | 0.14 | 0.90 | 9 | 0.18 | 0.87 | 62 | 0.31 | 0.68 | 7 | 0.53 | 0.60 |
| PCMCI+ | 21 | _0.43_ | 0.71 | 29 | 0.25 | 0.83 | 5 | 0.44 | 0.66 | _50_ | 0.32 | _0.57_ | 5 | _0.66_ | 0.50 |
| NOTEARS-MLP | 9 | 0.18 | _0.50_ | 21 | _0.32_ | _0.77_ | _4_ | 0.33 | 0.66 | 71 | 0.38 | 0.68 | 14 | 0.30 | 0.80 |
| NTS-NOTEARS | 17 | 0.32 | 0.75 | 18 | 0.18 | 0.84 | 6 | 0.40 | 0.71 | 53 | 0.10 | 0.76 | 5 | 0.54 | 0.50 |
| DAG-GNN | 13 | 0.31 | 0.70 | 17 | 0.11 | 0.90 | 7 | 0.22 | 0.83 | 66 | 0.21 | 0.76 | **3** | **0.75** | 0.37 |
| DYNOTEARS | 18 | 0.25 | 0.80 | 27 | 0.12 | 0.90 | 8 | 0.00 | 1.00 | 65 | 0.21 | 0.75 | _4_ | _0.66_ | 0.42 |
| CD-NOD | 10 | 0.37 | 0.57 | _16_ | 0.00 | 1.00 | 5 | _0.54_ | _0.62_ | 54 | 0.25 | 0.63 | _4_ | 0.50 | _0.33_ |
| LIN | 61 | 0.21 | 0.88 | 98 | 0.13 | 0.93 | _4_ | 0.33 | 0.66 | 56 | 0.31 | 0.63 | _4_ | 0.60 | 0.40 |
| TCDF | 10 | 0.16 | 0.66 | **9** | 0.18 | **0.50** | 6 | 0.25 | 0.80 | 51 | 0.21 | 0.63 | 7 | 0.22 | 0.75 |
| CAUSALFORMER | 9 | 0.31 | 0.50 | 16 | 0.11 | 0.88 | 8 | 0.00 | 1.00 | 55 | _0.43_ | 0.59 | 10 | 0.00 | 1.00 |
| PROPOSED TTCD | **8** | **0.50** | **0.42** | **9** | **0.40** | **0.50** | **1** | **0.80** | **0.00** | **46** | **0.45** | **0.50** | **3** | _0.66_ | **0.25** |

quality causal graphs for non-stationary temporal data and can identify true causal edges with fewer spurious causal links compared to state-of-the-art baseline models.

## 6.1 ABLATION STUDY

A comparative study of the proposed framework and its different variants is performed to verify the effectiveness of each component in our framework. In the TTCD Normal Transformer variant, we used a standard transformer rather than a non-stationary transformer, keeping the Causal Structure Learner unchanged. The TTCD w/o DSB variant removes the de-stationary factor learning block (DSB) to evaluate its contribution, and TTCD w/o Frequency excludes the frequency-domain attention from non-stationary transformer while keeping other modules unchanged. The evaluation results in Table 7 show that the non-stationary transformer learns informative latent features better than a standard transformer on multivariate non-stationary data. Moreover, removing either DSB or frequency domain attention block degrades performance across all datasets, demonstrating their effectiveness for robust causal discovery.

## 7 CONCLUSION

In this paper, we propose TTCD, a score-based causal structure learning method for non-stationary time series data that integrates the non-stationary transformer and a custom Causal Conv2D module. The proposed method leverages the temporal and frequency domain attentions enhanced by non-stationary profiling and de-stationary factor learning networks to learn important non-stationary features and refined reconstructed signals. The custom causal structure learner keeps the causal contributor of each target/effect variable isolated from other target variables, which helps to estimate a better causal structure from distilled signals. Unlike many existing methods, the proposed framework does not require prior knowledge about variable independence, noise distribution, or the underlying data generation process. We conducted extensive experiments on synthetic, benchmark, and real world complex time series datasets to demonstrate the performance of the proposed causal discovery framework. Experimental analysis demonstrates that the TTCD framework achieves superior causal graph learning capability compared to state-of-the-art baselines. In the future, we will analyze more benchmarks and real world datasets from other domains and evaluate the sensitivity of different parameters.

Table 2: Ablation analysis between the proposed framework and its different variants.

| DATASET | TTCD | | | TTCD W/O DSB | | | TTCD W/O FREQUENCY | | | TTCD NORMAL XFORMER | | |
|---|---|---|---|---|---|---|---|---|---|---|---|---|
| | SHD↓ | F1↑ | FDR↓ | SHD↓ | F1↑ | FDR↓ | SHD↓ | F1↑ | FDR↓ | SHD↓ | F1↑ | FDR↓ |
| DATASET-1 | 8 | 0.50 | 0.42 | 10 | 0.28 | 0.60 | 12 | 0.25 | 0.71 | 13 | 0.13 | 0.83 |
| DATASET-2 | 9 | 0.40 | 0.50 | 11 | 0.15 | 0.75 | 12 | 0.14 | 0.80 | 14 | 0.13 | 0.77 |
| TKE | 1 | 0.80 | 0.00 | 6 4 | 0.50 | 0.60 | 4 | 0.33 | 0.66 | 14 | 0.22 | 0.77 |
| ARCTIC SEA ICE | 46 | 0.45 | 0.50 | 63 | 0.34 | 0.66 | 50 | 0.46 | 0.54 | 63 | 0.35 | 0.66 |
| FMRI | 3 | 0.66 | 0.25 | 6 | 0.36 | 0.66 | 7 | 0.36 | 0.66 | 7 | 0.22 | 0.75 |

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

# Appendix

## A  COMPARISON OF CAUSAL DISCOVERY METHODS

Causal discovery methods consider different assumptions about data distribution and causal structure of the target system. A comprehensive summary of the assumptions used in different causal discovery methods is provided in Table 3. The four most commonly used assumptions are mentioned in the columns: acyclicity, stationary or non-stationary data, Markov & Faithfulness assumption, and causal sufficiency. The last column of the table mentions any specific criteria used by the method.

Table 3: Assumptions used by different existing causal discovery methods.

| METHOD | ACYCLICITY | STATIONARITY | MARKOV & FAITHFULNESS | CAUSAL SUFFICIENCY | OTHERS |
|---|---|---|---|---|---|
| GRANGER CAUSALITY | No | YES | No | YES | LINEAR RELATIONSHIP |
| PC-STABLE COLOMBO ET AL. (2014) | YES | YES | YES | YES | |
| PCMCI RUNGE ET AL. (2019B) | YES | YES | YES | YES | |
| PCMCI+ RUNGE (2020) | YES | YES | YES | YES | |
| NOTEARS-MLP ZHENG ET AL. (2020) | YES | No | YES | YES | LINEAR RELATIONSHIP |
| NTS-NOTEARS SUN ET AL. (2023) | YES | YES | YES | YES | NONLINEAR RELATIONSHIP |
| DYNOTEARS PAMFIL ET AL. (2020) | YES | YES | YES | YES | |
| TCDF NAUTA ET AL. (2019) | YES | YES | No | YES | ATTENTION WEIGHTS CAPTURE CAUSAL IMPORTANCE |
| CD-NOD HUANG ET AL. (2020) | YES | No | YES | No | DISTRIBUTION SHIFTS REVEAL CAUSAL INFLUENCES |
| TS-FCI ENTNER & HOYER (2010) | No | YES | YES | YES | |
| VAR-LINGAM HYVÄRINEN ET AL. (2010) | YES | YES | No | YES | LINEAR RELATIONSHIP |
| CAUSALFORMER KONG ET AL. (2024) | YES | YES | No | YES | |
| SPACETIME MAMECHE ET AL. (2025) | YES | No | YES | YES | DISTRIBUTION SHIFT, INDEPENDENT CHANGES |
| LIN LIU & KUANG (2023) | YES | No | YES | YES | INTERVENTIONAL DATA AND EQUIVALENCE CLASS |
| TTCD (OURS) | YES | No | YES | YES | NONLINEAR RELATIONSHIP |

## B  IDENTIFIABILITY OF CAUSAL GRAPH

Considering the assumptions stated earlier and the given time series follows a nonlinear function with additive noise, the full-time causal graph $G$ is identifiable from data distribution. This renders equation 1 follows an identifiable functional model class (IFMOC) Peters et al. (2011; 2013) where the causal graph is acyclic. Motivated by Peters et al. (2013) we derived the following explanation of identifiability. Assume we got two different directed acyclic causal graphs $G_1$ and $G_2$ from the distribution of $X_t$. Suppose an edge between $x^i$ and $x^j$ with a time lag $p$, $x_{t-p}^i \to y_t^j$ which exist in $G_1$ but not in $G_2$. Based on causal faithfulness assumption, from $G_1$ we have $x_{t-p}^i \not\perp y_t^j | \{X_{t-l}^k \setminus \{x_{t-p}^i, y_t^j\}, k \in n, 1 \leq l \leq l_{max}\}$. Similarly, the Markov condition on $G_2$ provides $x_{t-p}^i \perp y_t^j | \{X_{t-l}^k \setminus \{x_{t-p}^i, y_t^j\}, k \in n, 1 \leq l \leq l_{max}\}$. This creates a contradiction in data distribution, hence the full-time causal graphs $G_1$ and $G_2$ must be equal and represent the same IFMOC.

## C    SYNTHETIC DATASET AND GROUND TRUTH CAUSAL GRAPH

The synthetic dataset-1 is generated using the following equations and the noise signals used in this dataset are generated by the Gaussian distribution. Here we used sinusoidal nonlinearity and this dataset represents both instantaneous and time-lagged causal relationships.

$$X_t^1 = 0.5X_{t-5}^1 + 0.5X_{t-2}^1 + \varepsilon_1$$

$$X_t^2 = 0.1X_t^1 + 0.7X_{t-1}^1 + 1.5sin(t/50) + \varepsilon_2$$

$$X_t^3 = 0.8X_{t-1}^1 + \varepsilon_3$$

$$X_t^4 = 0.2X_{t-1}^4 + 0.4X_t^3 + 0.4X_{t-1}^3 + 0.4X_{t-1}^1 +$$

$$sin(\frac{t}{50}) + sin(\frac{t}{20}) + \varepsilon_4$$

The synthetic dataset-2 is generated using the equations given below, and the noise signals used in this dataset are generated by the Poisson distribution. Here we used the exponential non-linearity using the term $f(x) = x + 5x^2 e^{-\frac{x^2}{20}}$. All the variables of this dataset are also non-stationary.

$$X_t^1 = \frac{t + 0.2t}{300}$$

$$X_t^2 = 0.2f(X_{t-1}^2) + 0.3f(X_{t-1}^1) + \mathcal{N}(0,1)$$

$$X_t^3 = 0.5f(X_{t-1}^3) + 0.2f(X_{t-4}^1) + \mathcal{N}(0,1)$$

$$X_t^4 = 0.7f(X_{t-1}^4) + 0.5f(X_{t-3}^3) + 0.8f(X_t^2) + \mathcal{N}(0,1)$$

$$X_t^5 = 0.6f(X_{t-2}^5) + 0.2f(X_{t-1}^1) + \mathcal{N}(0,1)$$

The ground truth causal graph of the synthetic dataset-1 is illustrated in Figure 3a. Where X1 is a common cause of all other variables. The time lag between each cause and effect variable pair is provided on the edge connecting them. Figure 3b visualizes the true causal relationships between different variables of the TKE dataset.

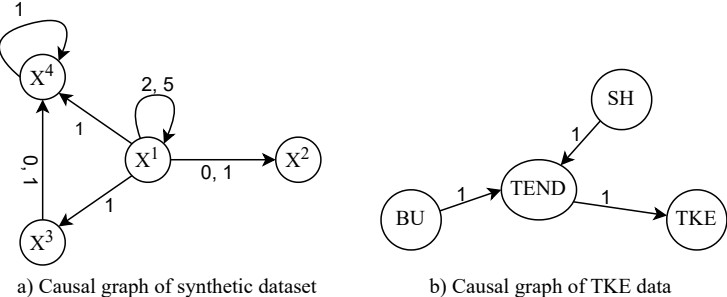

a) Causal graph of synthetic dataset          b) Causal graph of TKE data

Figure 3: Causal graph of (a) our synthetic dataset-1 and (b) the real world Turbulence Kinetic Energy (TKE) dataset.

## D    ARCTIC SEA ICE DATA

The following 11 atmospheric variables with the sea ice extent are included in the Arctic Sea Ice Dataset. This time series data contains monthly averages from 1980 to 2018 over the Arctic region of 60N.

Table 4: Variables in the Arctic Sea Ice Data.

| ABBREVIATION | FULL NAME | STATIONARY |
|---|---|---|
| HFLX | HEAT FLUX | NO |
| CC | CLOUD COVER | NO |
| SW | NET SHORTWAVE FLUX | YES |
| U10M | ZONAL WIND AT 10M | YES |
| SLP | SEA LEVEL PRESSURE | YES |
| PRE | TOTAL PRECIPITATION | YES |
| ICE | SEA ICE | NO |
| LW | NET LONGWAVE FLUX | NO |
| V10M | MERIDIONAL WIND AT 10M | YES |
| CW | TOTAL CLOUD WATE PATH | YES |
| GH | GEOPOTENTIAL HEIGHT | YES |
| RH | RELATIVE HUMIDITY | NO |

Table 5: Non-stationarity test results for variables of the TKE dataset.

| VARIABLES | ADF TEST | | KSSP TEST | |
|---|---|---|---|---|
| | P-VALUE | STATIONARY | P-VALUE | STATIONARY |
| SH | 0.32 | NO | 0.01 | NO |
| BU | 0.72 | NO | 0.01 | NO |
| TEND | 0.66 | NO | 0.01 | NO |
| TKE | 0.58 | NO | 0.01 | NO |

## E  NON-STATIONARITY TEST RESULTS FOR REAL WORLD DATASETS

The non-stationarity feature of the real world TKE and Arctic Sea Ice datasets is evaluated using the Augmented Dickey–Fuller test (ADF) Cheung & Lai (1995) and Kwiatkowski-Phillips-Schmidt-Shin test (KPSS) Kwiatkowski et al. (1992) statistical test methods for time series data. The ADF test method assumes a null hypothesis: the time series has a unit root and is not stationary. Then try to reject the null hypothesis and if failed to be rejected, it suggests the time series is not stationarity. For the ADF test, if the p-value of a time series is higher than the 0.05 alpha level the null hypothesis cannot be rejected. So the time series is not stationary. The KPSS test works in a somewhat similar manner to the ADF test but assumes an inverse null hypothesis. The null hypothesis of the KPSS method is that the time series is stationary. If the p-value is less than 0.05 alpha level, we can reject the null hypothesis and derive that the time series is not stationary.

The statistical non-stationarity test results for the TKE dataset are given in Table 5 and the results for the Arctic Sea Ice data are available in Table 6. The non-stationarity test results revealed that the TKE dataset contains only non-stationary variables, and both test methods have agreement on the test outcome. For Arctic Sea Ice dataset, the ADF test found 4 non-stationary variables; on the other hand KSSP method found 3 non-stationary variables. Therefore, we can say that the Arctic Sea Ice data have a mixture of both non-stationary and stationary variables.

## F  HYPERPARAMETERS

To find the best hyperparameters for baseline methods, we started using the parameters suggested by the authors and gradually tuned those values to obtain better evaluation results. The results reported in the comparative analysis of the main article are obtained with tuned hyperparameters. The parameters used to generate evaluation results are given here.

- PCMCI: Conditional Independence Test = ParCorr, $tau\_max$ = Maximum time lag, $pc\_alpha$ = None [So the model will use the optimal value from the list $\{0.05, 0.1, 0.2, 0.3, 0.4, 0.5\}$], $alpha\_level$ = 0.01

- PCMCI+: Conditional Independence Test = ParCorr, $tau\_max$ = Maximum time lag, $pc\_alpha$ = None [So the model will use the optimal value from the list $\{0.001, 0.005, 0.01, 0.025, 0.05\}$]

Table 6: Non-stationarity test results for variables of the Arctic Sea Ice dataset.

| VARIABLES | ADF TEST | | KSSP TEST | |
|---|---|---|---|---|
| | P-VALUE | STATIONARY | P-VALUE | STATIONARY |
| HFLX | 0.10 | **No** | 0.07 | **Yes** |
| CC | 0.00 | **Yes** | 0.02 | **No** |
| SW | 0.00 | Yes | 0.10 | Yes |
| U10M | 0.00 | Yes | 0.10 | Yes |
| SLP | 0.00 | Yes | 0.10 | Yes |
| PRE | 0.00 | Yes | 0.09 | Yes |
| ICE | 0.75 | **No** | 0.01 | **No** |
| LW | 0.28 | **No** | 0.10 | **Yes** |
| V10M | 0.00 | Yes | 0.10 | Yes |
| CW | 0.00 | Yes | 0.10 | Yes |
| GH | 0.01 | Yes | 0.10 | Yes |
| RH | 0.11 | **No** | 0.01 | **No** |

- NOTEARS-MLP: lambda1 = 0.01, lambda2 = 0.01, rho = 1.0, alpha = 0.0, $w\_threshold$ = 0.3

- NTS-NOTEARS: lambda1 = 0.0005, lambda2 = 0.001, $w\_threshold$ = 0.3, rho = 1.0, alpha = 0.0, $number\_of\_lags$ = Maximum time lag

- DYNOTEARS: $tau\_max$ = Maximum time lag, $w\_threshold$ = 0.01, $lambda\_w$ = 0.05, $lambda\_a$ = 0.05

- CD-NOD: indep_test = fisherz

- LIN: E(no of intervention)=1, $no\_hidden\_layer$= [1, 2], $hidden\_dim$= [3, 4]

- Proposed Method: lambda1 = 0.9, alpha=1.0, rho = 1.0 $w\_threshold$ = 0.002, 0.004, 0.007, 0.17

## G ABLATION STUDY

To understand the effectiveness of the proposed non-stationary transformer with the custom Causal Conv2D module, we created one variant of the proposed framework without using a transformer. The architecture of this model variant is illustrated in Figure 4. Here, we replaced the transformer from the Non-Stationary Feature Learner with a comparatively simple convolutional neural network module. The module learns latent temporal features of the input data and integrates the factors learned by the de-stationary factor learning MLP. Finally, these rescaled latent features are provided to the Causal Structure Learner module to generate the causal graph. To optimize the Causal Conv2D model, we used three components of the combined loss function: target estimation loss ($L_{te}$), acyclicity constraint, and $L1$ regularization of the learned adjacency matrix. The input data reconstruction loss is not included in the objective function, as we did not use an autoencoder architecture.

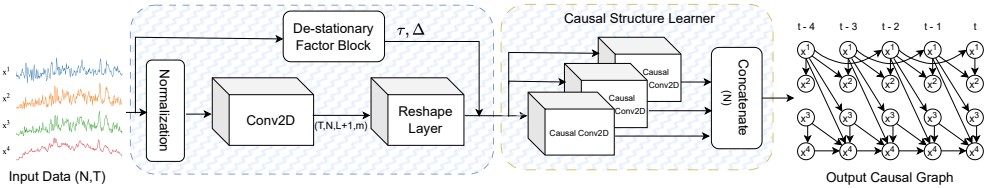

Figure 4: The structure of Causal Conv2D model without the transformer. Instead of using a non-stationary transformer, a Conv2D block is used to learn non-stationary features with the de-stationary factor MLP.

For TTCD Causal Conv1D, we used a 1D variant of the proposed custom Causal Conv2D layer. To incorporate this Conv1D layer into the model, we flattened the latent representation generated

by non-stationary transformer of the proposed architecture. The same training and optimization process was utilized for both models. From the experimental analysis of these ablation models, we can see that the non-stationary transformer and Custom causal Conv2D layer improve the causal graph learning performance of the proposed model with a significant margin for each evaluation score.

Table 7: Ablation study on normal transformer (TTCD-Without-Transformer) and 1D design of the causal structure learner (TTCD-Causal-Conv1D).

| DATASET | PROPOSED TTCD | | | TTCD-CAUSAL-CONV1D | | | TTCD-WITHOUT-TRANSFORMER | | |
|---|---|---|---|---|---|---|---|---|---|
| | SHD↓ | F1↑ | FDR↓ | SHD↓ | F1↑ | FDR↓ | SHD↓ | F1↑ | FDR↓ |
| DATASET-1 | 8 | 0.50 | 0.42 | 15 | 0.12 | 0.87 | 12 | 0.33 | 0.66 |
| DATASET-2 | 9 | 0.40 | 0.50 | 10 | 0.11 | 0.90 | 10 | 0.16 | 0.66 |
| TKE | 1 | 0.80 | 0.00 | 7 | 0.22 | 0.83 | 5 | 0.44 | 0.66 |
| ARCTIC SEA ICE | 46 | 0.45 | 0.50 | 59 | 0.39 | 0.63 | 61 | 0.39 | 0.63 |
| FMRI | 3 | 0.66 | 0.25 | 8 | 0.20 | 0.80 | | | |

