# OpenReview forum: "TTCD: Transformer Integrated Temporal Causal Discovery from Non-Stationary Time Series Data"
_ICLR.cc/2026/Conference — ICLR 2026 Conference Withdrawn Submission_

### Official Review · Reviewer_tUz8 · 2025-10-15

**Soundness:** 2
**Presentation:** 2
**Contribution:** 3
**Rating:** 4
**Confidence:** 4

**Summary:**

This paper introduces a novel architecture for causal discovery from nonstationary time series data. The problem is both important and challenging, and the proposed architectural approach is interesting. However, the submission in its current form is not ready for publication at ICLR. The primary concerns are an underdeveloped empirical evaluation that lacks the necessary rigor and detail to support the paper's claims. Additionally, the paper suffers from some clarity issues regarding the model's architecture and occasional formatting problems. For the paper to be considered for acceptance, these major issues must be thoroughly addressed.

**Strengths:**

*   The paper tackles the challenging and timely problem of causal discovery in nonstationary time series.
*   The proposed architecture is novel and presents an interesting conceptual approach to handling nonstationarity when identifying causal relationships.

**Weaknesses:**

My review identifies three main areas for improvement: the empirical evaluation, architectural clarity, and overall presentation.

**1. Insufficient and Unclear Empirical Evaluation (Major Weakness):**

In the absence of theoretical support, the burden falls on the empirical evaluation to be exceptionally thorough and convincing.
However, the empirical evaluation is the most critical weakness of the paper and is not currently at the standard required for ICLR.

*   **Lack of Reproducibility and Detail:** The methodology for hyperparameter selection is opaque. The authors state that baseline hyperparameters were "carefully selected," which is insufficient. Neither the process for this selection nor the search space is described. Critically, there is no information provided on how hyperparameters for the proposed model were chosen (e.g., Number of Layers, Optimizer, and encoding dimensions), which prevents reproducibility and fair comparison.
*   **Limited Scale and Scope:** The evaluation is conducted on only five individual time series. This is too limited a sample to draw generalizable conclusions about the method's effectiveness, as it does not allow for robust performance metrics and standard deviations. Especially for synthetic data, the reason to evaluate on only 2 SCMs is unclear.
*   **Training stability:**  How stable is the proposed approach across training runs? The reported results appear mixed, with the proposed method sometimes performing strikingly better and other times not. Is this consistent between runs?
*   **Ambiguous Ground Truth:** For the real-world datasets (Arctic sea ice, fMRI), the ground truth causal relationships are not specified or justified. This makes it impossible for a reader to independently assess the correctness of the discovered graphs.
*   **Missed Opportunity for Standardized Benchmarking:** The authors are encouraged to consider evaluating their method on recent established public benchmarks for causal discovery from time series [1, 2, 3]. This would enable a more rigorous, standardized, and directly comparable evaluation against a wider range of methods and allow for proper quantification of robustness.

**2. Architectural Ambiguity and Lack of Clarity:**

Several key aspects of the proposed architecture and its motivation are not clearly explained.

*   **Vague Figures:** Figure 1, which depicts the architecture, is ambiguous and underutilized. The specific function of the "Decoder layers" is not labeled and can only be inferred from the text. Given the white space, these components could be further annotated.

*   **Unexplained Design Choices:** Key design choices lack justification. For instance:
    *   How are the three separate input paths to the decoder combined or processed?
    *   What is the motivation for using two distinct loss terms related to the predicted `X` at different stages of the model? A clear ablation study or theoretical justification for this design is needed.
    *   Basic architectural details, such as the number of layers deployed in each component, are missing.

**3. Presentation and Formatting Issues:**

The manuscript suffers from several formatting and presentation issues that detract from its professionalism and readability.

*   **Typesetting:** The tables frequently overflow the page margins, both vertically and horizontally. Figure 1 appears to have excessive white space that could be used more effectively.
*   **Missing Citations:** Key claims or statements (e.g., in Line 96 and Line 125) lack necessary citations to support them.
*   **Broken References:** A reference on Page 2 breaks to page 3.
*   **Lack of Precision:** The core assumptions of the model (e.g., "The general assumption") should be stated with greater mathematical precision.

**Questions:**

Some Questions that could be discussed:

1.  Could you elaborate on the theoretical or intuitive connection between the concept of "Causal Consistency" used in your model and the property of nonstationarity in the data?
2.  What are the data requirements for your architecture to perform well? For instance, how does its performance scale with the length of the time series, the number of variables, or the degree of nonstationarity?

---

### Official Review · Reviewer_iBkv · 2025-10-29

**Soundness:** 2
**Presentation:** 3
**Contribution:** 2
**Rating:** 4
**Confidence:** 3

**Summary:**

The paper proposes TTCD, an end-to-end framework for discovering both contemporaneous and lagged causal relationships from non-stationary, nonlinear, and noisy multivariate time series.
The the Non-Stationary Feature Learner integrates temporal and frequency-domain attention within a transformer architecture. And the Causal Structure Learner employs a custom Causal Conv2D module to simultaneously learn lagged and instantaneous causal links. Empirical validation are conducted on several real-world datasets.

**Strengths:**

S1: An end-to-end solution. A novel end-to-end framework that integrates transformer-based representation learning with causal discovery specifically tailored for non-stationary time series. It's a setting where most existing methods struggle or make restrictive assumptions. And the design of Non-Stationary Feature Learner make sense.

S2: The paper is clearly written and well-motivated.

**Weaknesses:**

W1: Limited theoretical justification for causal discovery. The reconstruction-guided causal signal distillation relies on the assumption that the transformer decoder's reconstructed output better preserves true causal signals while suppressing noise and spurious correlations. However, no theoretical or empirical analysis is provided to validate this key claim.
Analysis regarding this issue is expected. I would like to reconsider the soundness score if this issue can be solved.

W2: Incomplete experimental reporting. The paper lacks runtime efficiency analysis and reports results without averaging over multiple runs, reducing reliability and reproducibility.

W3: Thresholding strategy for edge selection is ad-hoc. The final causal graph is obtained by thresholding learned weights, but the choice of the threshold is not principled. It appears to be tuned per dataset, which undermines reproducibility and practical deployment.

W4: Several works regarding causal discovery under non-stationarity are missed. For example, Regime-PCMCI, CDANs, and CR-VAE. These works are expected to be evaluated or discussed. I would like to reconsider the contribution score if this gap can be addressed.

**Questions:**

As with the weakness above.

---

### Official Review · Reviewer_PGPu · 2025-10-29

**Soundness:** 2
**Presentation:** 2
**Contribution:** 2
**Rating:** 4
**Confidence:** 4

**Summary:**

The paper proposes TTCD, an end-to-end framework for temporal causal discovery on non-stationary, nonlinear, noisy time series. The method is a non-stationary transformer that integrates temporal and frequency features, revealing temporal causal structure with a de-stationary factor block and a causal structure learner. Experiments cover a wide range of baselines on two synthetic datasets, an fMRI benchmark, and two Earth-system datasets.

**Strengths:**

- The proposed non-stationary transformer integrates both temporal and frequency features, providing a richer analysis on time series
- The empirical evaluation covers a wide range of methods and datasets.

**Weaknesses:**

- The paper needs a detailed elaboration on its research target. In the beginning of the paper, it states that the paper proposed a causal discovery method "without any noise or data distribution assumptions". While appreciating the ambition, could you provide more demonstration on how the proposed methods unify or relax the assumptions for identifiability? For example, consider the following time series: $$X_t \mid Y_t=y \sim \mathcal{N}(y,1) \quad Y_t \sim\mathcal{N}(0,1)$$ with the true causal graph $Y_t \rightarrow X_t$.  This toy example satisfies the assumptions listed in section 3, however, to the best of my knowledge, it is not identifiable. Therefore, some inherent assumptions, perhaps about the non-stationarity, are missing in the current manuscript.
- The hyper-parameter specification may be complex. The number of parameters can be significantly large when stacking more layers in the decoder and encoder. It would be much better to provide an ablations on the layers, kernel size, etc..
- Some minor concerns on baselines.
	- For synthetic dataset 1, it can be seen as applying (soft) interventions on $X^2_t$ and $X^4_t$.  For synthetic dataset 2, it can be seen as applying (soft) interventions on $X^1_t$. Therefore, setting $E=1$ may not be suitable for the LIN method.
	- For time-lag methods, they inherently employ a prior between variables. For example, $X_t \rightarrow X_{t-1}$ is impossible. Therefore, it may be better to also give such prior to the NOTEARS-MLP method and DAG-GNN method.

**Questions:**

- How do frequency features benefit causal discovery in the proposed method?
- What are the formal assumptions on the non-stationarity?
- In practice, how can we choose the hyperparameters of the proposed methods? Even a heuristic way?

---

### Official Review · Reviewer_X2mc · 2025-11-02

**Soundness:** 2
**Presentation:** 2
**Contribution:** 2
**Rating:** 2
**Confidence:** 5

**Summary:**

The paper proposes a novel score-based method for causal structure learning in nonstationary time series. It integrates a nonstationary Transformer with a custom Causal Conv2D module. The model uses temporal and frequency-domain attention, enhanced by nonstationary profiling and de-stationary factor networks, to capture local and global nonstationary features. A causal structure learner then infers the causal relations for each variable at each layer. The approach models both contemporaneous and lagged effects.

**Strengths:**

1. The paper tackles a practical challenge in time-series causal discovery: many real-world series are nonstationary.

2. The model is validated on synthetic and real-world datasets and generally achieves higher accuracy than the baselines.

3. The framework of the model is clear and well-motivated.

4. The ablation study validates the necessity of each module in the model.

**Weaknesses:**

1. There is no theoretical guarantee. Although the paper discusses identifiability of the causal graph, the discussion does not align with the proposed model. There is a gap between the IFMOC framework (Peters et al., 2011; 2013) and what is assumed here. Specifically, the main paper does not state that its applicability is restricted to additive-noise models, and it does not apply assumptions on the data-generating process and on the noise distribution. In Definition 1, it does not require independence between the noise term and the variables, nor i.i.d. noise. A further concern is nonstationarity: Peters et al. (2013) note that some results require stationarity and/or α-mixing or geometric ergodicity (see, e.g., Chu and Glymour, 2008).

2. The novelty of each module is unclear. Beyond the Causal Structure Learner, which components are modified relative to prior work, and which are used as is? For the new Causal Structure Learner, since it estimates causal strengths directly, it would be helpful to report those learned strengths in the experiments and compare them with the ground truth.

3. Important model details are missing, including how the adjacency threshold is chosen and how the sparsity loss coefficient is set.

4. The experimental section should include multiple trials with standard errors. Please also report scalability, computational complexity, and running time.

Some minor comments:

1. Definition 1 could be clearer about what is being defined. In addition, the definition of $E$ appears to be incorrect.

**Questions:**

1. What is the difference between Eq. 7 and the constraint used in DYNOTEARS?

2. What parameter configuration did you use for TCDF?

3. Did you perform any tests for nonstationarity on the fMRI datasets?

---

### Note · Authors · 2025-11-24

**Comment:**

We are thankful to all respected reviewers for their valuable and insightful suggestions and questions. These reviews will help us to further enhance the theoretical analysis and empirical evaluation of the proposed model to improve its quality. We will work on the points identified by the reviewers and try with a better presentation in the future.

**Withdrawal Confirmation:**

I have read and agree with the venue's withdrawal policy on behalf of myself and my co-authors.